# Effects of Different Exercise Interventions on Fall Risk and Gait Parameters in Frail Patients After Open Heart Surgery: A Pilot Study

**DOI:** 10.3390/medicina61020206

**Published:** 2025-01-24

**Authors:** Vitalija Stonkuvienė, Raimondas Kubilius, Eglė Lendraitienė

**Affiliations:** Department of Rehabilitation, Faculty of Nursing, Medical Academy, Lithuanian University of Health Sciences, 44307 Kaunas, Lithuania; raimondas.kubilius@lsmuni.lt (R.K.); egle.lendraitiene@lsmu.lt (E.L.)

**Keywords:** frailty, cardiac rehabilitation, open heart surgery, computer-based interactive program

## Abstract

*Background and Objectives*: Research on the effectiveness of different exercise programs on reducing fall risk and improving gait parameters among frail patients after open heart surgery is scarce. Therefore, the aim of this study was to evaluate and compare the effects of different exercise interventions on fall risk and gait parameters in frail patients after open heart surgery during inpatient rehabilitation. *Materials and Methods*: A prospective randomized study was conducted at Kulautuva Hospital of Rehabilitation, Hospital of Lithuanian University of Health Sciences Kauno Klinikos, from July 2021 to November 2023. A total of 105 pre-frail and frail patients were randomized into three groups: control (CG, n = 35), intervention 1 (IG-1, n = 35), and intervention 2 (IG-2, n = 35). All three groups completed a conventional rehabilitation program that included aerobic training tailored based on each person’s health status six times/week. The IG-1 additionally received multicomponent dynamic aerobic balance and strength training three times/week, and the IG-2 received a combined computer-based interactive program three times/week. The overall stability index, as an outcome of fall risk, Short Physical Performance Battery (SPPB) score, and gait parameters (geometry, stance, and timing) were assessed before and after rehabilitation. *Results*: Of the 105 patients, 87 completed the study. The median age of the patients was 71 years (range: 65–88); 64.76% were male. After rehabilitation, within-group comparisons showed a significant improvement in the overall stability index, SPPB, and all phase gait parameters in all groups. Of all geometry gait parameters, none of the groups showed a significant improvement in step width, and foot rotation was significantly reduced only in the IG-2. All timing gait parameters improved in the CG and IG-2 after rehabilitation, while in the IG-1, only gait speed was significantly improved. Between-group comparisons after rehabilitation revealed significant differences in the overall stability index among the groups and in all phase gait parameters except for stance phase between the IG-1 and the IG-2. The greatest significant difference was documented for the double stance phase between the IG-1 and the IG-2, and the smallest was recorded for the single limb support phase. Significantly greater differences in step time and stride time were found in the IG-1 and the IG-2 than in the CG. *Conclusions*: All applied interventions were effective in reducing fall risk based on the overall stability index. However, interactive exercise interventions decreased fall risk based on the overall stability index by one-fifth as compared to the conventional rehabilitation program. The incorporation of interactive exercise interventions in rehabilitation resulted in improved double stance phase, stride time, and step time parameters, but did not have any effect on gait speed as compared to other exercise interventions.

## 1. Introduction

Exercise-based cardiac rehabilitation (CR) is widely recommended for patients after cardiac events [1]. CR is considered the standard of care to facilitate recovery from cardiovascular disease (CVD), using exercise therapy, management of nutrition, modification of risk factors, psychosocial support, and patient education. It is known that functional improvement occurs through rehabilitation exercises, education, and training. Exercise-based CR increases VO_2_ max and endurance, and contributes to reduced mortality and rehospitalization [2]. Research shows that the benefits of CR can extend beyond improvements in the secondary prevention of CVD to alleviate age-related health deficits [3]. Elderly patients are affected by CVD, and with an aging population, the number of elderly people with CVD will increase dramatically [4,5]. There is a strong bidirectional association between CVD and frailty, and frail older adults are at a greater risk of developing CVD [6].

Frailty is most commonly defined as an age-related biological syndrome characterized by a decreased biological reserve due to impairments in the regulation of multiple physiologic systems [7]. This leads to increased vulnerability of an individual to physiologic stressors, and it is linked to adverse outcomes such as disability, hospitalization, or death [8]. There are several frailty phases, ranging from the complete absence of frailty to vulnerable state, followed by physical frailty. Vulnerable state, or pre-frailty, can increase the risk of developing frailty [9]. According to one meta-analysis, the overall prevalence of frailty ranges from 40% to 50% depending on the frailty tool used [3]. Frailty is closely related to falls, and it is the second leading unintentional cause of injury and death among older adults [10]. Falls, as one of the major “geriatric syndromes”, like the walking process, are the result of neurological, musculoskeletal, nutritional, and cardiovascular integration [11]. Open heart surgery, which is frequently performed among patients with frailty, can cause physiological stress. Therefore, 28% of elderly people arriving at inpatient rehabilitation already have decreased skeletal muscle strength and physical performance, and this has a huge impact on balance and general performance in carrying out daily activities [12], including walking [13]. Exercise-based CR sets a goal to improve fall risk and gait parameters in frail patients after open heart surgery. There are no universally accepted and evaluated standards for exercise-based CR regarding its contents, frequency, time, and intensity for frail patients after cardiac intervention [1]. Traditionally, conventional training means are used for balance and gait training in frail patients during rehabilitation after open heart surgery. Research indicates the benefits of different interventions such as multicomponent programs [14] or interactive programs [15].

Based on the literature, a multicomponent program should include elements of aerobics, strength, balance, and flexibility. These elements improve gait and reduce the risk of falling [16]. Studies have shown that frailty patients are recommended a combination of aerobics and strength training that increases muscle mass and improves strength, function, and balance. Muscle strength and balance training are key components of exercise programs aimed at reducing frailty and fall risk scores in older adults [14,16].

In computer-based rehabilitation, pressure platforms and video cameras are used for computerized balance and gait analysis, which allow for the objective determination of balance and gait parameters as well as the evaluation of the effectiveness of the applied program [17]. Interactive, computerized, physical performance-enhancing programs that are safe can improve balance, reduce physical impairment, and combine real-time motion detection and feedback to the user. They interact with virtual objects and evaluate movement accuracy, speed, and activity duration [15]. These interventions improve lower-limb muscle strength, walking speed, balance, and fall risk [15]. Computer-based interactive programs are useful, help motivate patients to exercise, and are increasingly being included in CR [18].

The application of multicomponent programs and interactive tools in rehabilitation of frail patients after open heart surgery is limited [15,19], and research on this topic is scarce. To our knowledge, there are no published articles that evaluated and compared the effects of multicomponent dynamic and combined computer-based interactive cardiac training programs. Therefore, the aim of this study was to evaluate and compare the effects of the conventional rehabilitation program and different complementary exercise programs (multicomponent dynamic and combined computer-based interactive cardiac training programs) on the fall risk and gait parameters of frail patients after open heart surgery during rehabilitation. We hypothesized that multicomponent dynamic and combined computer-based interactive cardiac training programs would reduce fall risk and improve gait parameters better than the conventional rehabilitation program.

## 2. Materials and Methods

### 2.1. Study Design, Setting, and Participant Selection

This prospective randomized experimental study was conducted at Kulautuva Hospital of Rehabilitation, Hospital of Lithuanian University of Health Sciences Kauno Klinikos, from July 2021 to November 2023. The Kauno Regional Biomedical Research Ethics Committee approved this study (No. BE-2-83, dated 22 July 2021). The study was carried out in accordance with the ethical principles outlined in the Declaration of Helsinki and was registered in a clinical trial registry at ClinicalTrials.gov (registration No. NCT06385041). 

All patients who arrived at Kulautuva Hospital of Rehabilitation and who had undergone open heart surgery—i.e., coronary artery bypass grafting, heart valve replacement, or complex surgery—and met the inclusion criteria were invited to participate in the study. Patients were included in the study if they met the following criteria:Older than 65 years;Vulnerable or frail (a score of ≥4 on the Edmonton Frail Scale, EFS);Underwent open heart surgery;Able to walk independently (without supportive aids);Walked ≥ 150 m based on the 6 min walk test;Signed an informed consent form.

The exclusion criteria were as follows:Refusal to participate in the study;Disorders and pathologies of the musculoskeletal system;Cognitive impairment;Severe underlying conditions (mental, visional, and hearing impairments, heart failure, severe anemia, complications of postoperative wound healing);Other acute conditions that could limit active participation in exercise programs.

### 2.2. Participant Assessment and Assignment

A total of 310 patients who underwent open heart surgery were admitted and registered at Kulautuva Hospital of Rehabilitation, Hospital of Lithuanian University of Health Sciences Kauno Klinikos. Of them, 205 patients did not meet the inclusion criteria. All patients who arrive at cardiac rehabilitation after open heart surgery are given a 20-day inpatient rehabilitation.

On the first day of arrival at cardiac rehabilitation, the candidates who met the inclusion criteria were introduced to the course of the study and received information about the study both verbally and in writing. After answering all the questions raised, as well as reading and signing an informed consent form, each patient received a unique code for data coding. After examination by a physical medicine and rehabilitation physician, each patient was given a rehabilitation plan.

On the second day of rehabilitation, patients were clinically examined and the assessment of frailty status, fall risk, and gait was performed. After this assessment, patients were randomly allocated to one of the three groups at a 1:1:1 ratio using the Research Randomizer program.

During the entire study period, all patients received therapeutic massage, physical factors, counseling by a psychologist and a social worker, medication treatment, management of risk factors, and a protein-enriched diet.

### 2.3. Study Interventions

Patients were allocated to the following three groups:Control group (CG), which only underwent a conventional rehabilitation program (Table 1, part A);Intervention group 1 (IG-1), which underwent the conventional rehabilitation program (Table 1, part A) plus the multicomponent dynamic training program (Table 1, part B);Intervention group 2 (IG-2), which underwent the conventional rehabilitation program (Table 1, part A) plus the combined computer-based interactive cardiac program (Table 1, part C).

Before starting the program, each participant underwent testing with a cycle ergometer to determine optimal parameters and to ensure a tailored physical workload. The cardiovascular stress test was performed using a cycle ergometer (Ergoline GmbH, Bitz, Germany). During this test, the electrocardiogram (ECG), heart rate, blood pressure, and the well-being of the subject were monitored, and all complaints were documented. The test started with a load of 25 W, which was gradually increased by 12.5 W every minute until the patient became exhausted or other criteria for study discontinuation were obvious: significant changes in ECG, hypotension (decrease in systolic BP > 20 mmHg), marked hypertension (>230/120 mmHg), chest pain, pronounced shortness of breath, or other exercise-limiting symptoms. The intensity of the achieved load was evaluated in W and metabolic units (MET). One MET is the amount of oxygen while sitting consumed per kg of body weight in 1 min and is equal to 3.5 mL/kg/min [20]. The cardiovascular stress test is the gold standard for the diagnostic evaluation of exercise intolerance, especially in cardiac patients, and for setting an individualized prescription for a structured exercise training program [21,22,23]. This test is typically performed on a treadmill or stationary bike. The test shows very good-to-excellent validity (r = 0.85–0.97) and moderate-to-high reliability (intraclass correlation coefficient = 0.85–0.91) [24].

### 2.4. Study Measures

All the patients were assessed twice: at the beginning of rehabilitation, i.e., on the second rehabilitation day (T0), and on the day before completing rehabilitation (T1). During both assessments, the same data were evaluated and collected, except for the EFS, which was applied only during T0 assessment, and the Borg rating of perceived exertion (RPE) scale, which was used during each exercise session [25].

The EFS was employed to evaluate frailty. This scale consists of 9 domains: cognition, general health status, functional independence, social support, medication use, nutrition, mood, continence, and functional performance. Each domain is assessed on a 0–2-point scale; the total maximal score available is 17. Based on the score obtained, the level of frailty is evaluated as follows: score of 0 to 3, fit; score of 4 to 5, vulnerable; score of 6 to 7, mild frailty; score of 8 to 9, moderate frailty; and score of 10 and more, severe frailty [26]. This biomedical study enrolled only those patients who were vulnerable and mildly-to-severely frail, i.e., patients who got a score of 4 and more.

Fall risk was assessed using the computerized Biodex Balance system SD v3.06 (Shirley, NY, USA). This system utilizes a multi-axial tilting platform, allowing for the assessment of patients’ fall risk and the comparison of the results in a certain age group. During dynamic fall risk assessment, the platform was set to permit up to 20° of deflection in all directions (level 8). Subjects were instructed to maintain an upright posture 3 times for 20 s, without falling and grasping the handrails, with their eyes open and feet shoulder-width apart. The subject applied pressure to the platform, and anterior, posterior, medial, and lateral sways were recorded. Then, a derivative overall stability index ranging from 0.0 (optimal balance/equivalent of low fall risk) to 5.0 (poor balance/equivalent of high fall risk) was generated [27].

Gait analysis was performed using a treadmill with an embedded pressure plate Zebris, FDM, GmbH, Bitz, Germany). During the study, the following 3 groups of parameters were evaluated: geometry (foot rotation, step length, stride length, step width), phase (stance, load response, single limb support, pre-swing, swing, double stance), and timing (step time, stride time, cadence, and gait speed). Patients were asked to walk barefoot at their preferred gait speed. Gait testing lasted for 30 s. The entire assessment was captured with two video cameras set up in the frontal and sagittal planes. Gait parameters were recorded in the software [28].

The Borg rating of perceived exertion (RPE) scale is the most widely used subjective tool to measure an individual’s effort and exertion during physical activity. Patients were asked to rate their exertion on a 6–20-point scale during exercises, combining all sensations of physical stress and fatigue. The patients’ heart rate, parameters of heart rhythm (extrasystoles), respiratory rate, sweating, and medications used (beta-blockers) were also assessed in parallel. In our study, we used the following scores: 9–11, corresponding to light physical activity; 12–13, moderate; 14–17, vigorous; and 18–20, very vigorous [25].

The Short Physical Performance Battery (SPPB) was used to evaluate the following three components: balance, gait speed, and timed chair stand. Each component was scored from 0 to 4, and the overall SPPB score, ranging from 0 to 12, was obtained by adding the scores of the individual components. A score of 0 indicated the worst performance, and a score of 12 indicated the best performance [29].

### 2.5. Statistical Analysis

The outcome measure SPPB was used to calculate a sample size. A target sample size of 105 (35 per group) was calculated to ensure an 80% power (β = 0.2) to detect a minimum significant change of 1 point in the SPPB (α = 0.05).

Statistical analysis was performed using Microsoft Excel 2019 and IBM SPSS Statistics 29.0 software. Data distribution was checked with the Shapiro–Wilk test. Categorical data were compared using the chi-square test; in the presence of a small number of cases, the exact Fisher test was applied. Categorical data are given as numbers with percentages. Normally distributed continuous data were reported as means with standard deviations (SD) and nonnormally distributed data as medians with ranges. Three independent groups were compared with one-way analysis of variance (ANOVA) when data were normally distributed and with the nonparametric Kruskal–Wallis test when data were nonnormally distributed. Where applicable, a Bonferroni criterion was used for the multiple comparisons procedure. Comparison of dependent groups, when data were nonnormally distributed, was performed with the nonparametric Wilcoxon test. The level of significance was set at *p* < 0.05.

## 3. Results

All 105 eligible patients were randomly assigned to three different groups: the control group (CG, n = 35), intervention group 1 (IG-1, n = 35), and intervention group 2 (IG-2, n = 35). A total of 87 patients completed the study: 26 patients in the CG, 29 in the IG-1, and 32 in the IG-2. The main reasons for withdrawal from the study were as follows: worsened health status (n = 11; four, four and three patients in the CG, IG-1, and IG-2, respectively), left home earlier (n = 5; three and two patients in the CG and IG-1, respectively), and declined to continue participation in the study (n = 2; two patients from the CG). The flowchart of the study is depicted in Figure 1.

Table 2 shows the baseline characteristics of the 105 participants. The patients’ groups were homogenous for age, sex, BMI, left ventricular ejection fraction, and surgery type (*p* > 0.05). The median age of the patients was 71 years (range, 65–88). Men accounted for 64.76% of the whole study population. The ANOVA test showed a significant difference in height (F(2) = 3.27, *p* = 0.042) and weight (F(2) = 4.32, *p* = 0.016) among the groups. Paired post hoc comparisons using the Bonferroni test revealed that the patients in the IG-1 had a significantly lower height and weight than their counterparts in the IG-2 (165 ± 7.84 vs. 169.91 ± 8.10, *p* = 0.040, and 72.36 ± 14.09 vs. 82.30 ± 13.52, *p* = 0.020, respectively). Coronary artery bypass grafting (57.14%) was the most common surgery performed before arrival at rehabilitation (Table 2).

As our study enrolled patients who all had a score of more than 4 on the EFS, the median EFS score of all patients was 6 (range, 4–11). Among all the patients, 43.81% were vulnerable, 39.05% had mild frailty, 10.48% had moderate frailty, and the remaining were severely frail.

### 3.1. Effectiveness of Cardiac Rehabilitation on Fall Risk

After testing patients with the Biodex Balance SD platform and evaluating the overall stability index, which reflects a patient’s fall risk, it was observed that during T0 assessment, there was no significant difference in the overall stability index among all groups (*p* > 0.05). During T1 assessment, comparison of the overall stability index among the groups with the Kruskal–Wallis test showed a significant difference (H(2) = 23.87; *p* < 0.001). Pair-wise comparisons between the CG and the IG-2 revealed that the overall stability index was smaller in the IG-2 by more than one fifth (22.73%; *p* < 0.001) and between the IG-1 and the IG-2, this parameter was smaller in the IG-2 by 15% (*p* = 0.004) (Table 3). Within-group comparisons showed a 15.38% decrease in the overall stability index between T0 and T1 assessments in the CG (*p* = 0.001): a 23.08% decrease in the IG-1 (*p* = 0.019); and a 26.09% decrease in the IG-2 (*p* < 0.001).

There was no significant difference in the SPPB score among the groups either during T0 or T1 assessment (*p* > 0.05). Within-group comparisons revealed significant differences in the SPPB score between T0 and T1 assessments in all three groups (*p* < 0.001).

### 3.2. Effectiveness of Cardiac Rehabilitation on Gait Parameters

Between-group comparisons showed that before rehabilitation, all three groups were not homogenous by all geometry and timing gait parameters (*p* < 0.05) but were homogeneous and did not differ significantly by all phase gait parameters (*p* > 0.05). After rehabilitation, based on the Kruskal–Wallis test results, significant differences in all phase gait parameters (*p* < 0.05), except for the stance phase (*p* = 0.063), were found only between the IG-1 and the IG-2. The greatest significant difference of 8.33% was documented between the IG-1 and the IG-2 during the double stance phase (*p* = 0.037), and the smallest significant difference of 4.51% was found during the single limb support phase (*p* = 0.043).

Within-group comparisons revealed that after rehabilitation, significant differences in all phase gait parameters were observed for all groups. Analyzing differences in geometry gait parameters within the groups, step length and stride length significantly increased in all groups (*p* < 0.01) and foot rotation significantly decreased only in the IG-2 (*p* = 0.048). Meanwhile, step width did not differ significantly between T0 and T1 assessments in all groups. For timing gait parameters, significant differences were found for all parameters in the CG and the IG-2 (*p* < 0.001), but in the IG-1, only gait speed increased significantly after rehabilitation (*p* < 0.001) (Table 4).

As geometry and timing gait parameters differed significantly among all three independent groups at T0 assessment (*p* < 0.05), to eliminate the lack of homogeneity, we calculated and compared these differences among the groups (Table 5). The results showed significant differences in step time and stride time between the CG and the IG-1 as well as between the CG and the IG-2 (*p* < 0.05). No significant differences in other gait parameters were found (*p* > 0.05).

## 4. Discussion

This study aimed at evaluating and comparing the effects of different exercise interventions on fall risk and gait parameters among vulnerable and frail patients after open heart surgery during cardiac rehabilitation. The results of this study revealed positive and statistically significant results in reducing the risk of falls and improving gait parameters with additional interventions in frail patients after open heart surgery.

Research shows that strength, power, and endurance training are the most used components for older adults [30]. It is widely known that multicomponent exercises are a recommended strategy for frail older adults that increases functional capacity, balance, strength, gait speed, cognitive function, and independence in the elderly [30,31]. We could not find any studies that used interactive technologies in patients with frailty, especially after open heart surgery. Studies conducted with interactive technologies in the field of frailty show that these exercises increase gait speed and balance [15]. Although the combined computer-based interactive cardiac program used in this study was superior compared to other programs, in the analysis of the literature, there is a lack of research on the effect of interactive training programs; therefore, more studies are required to validate the effects of interactive exercise training [15]. As a result, it was difficult to perform a comparable analysis with other research in this field, because we could not find similar studies investigating multicomponent and computer-based exercises programs which were used together.

Cadore et al. noted that multicomponent training is one of the most effective ways of improving physical fitness, gait ability, balance, and strength performance and reducing the rate of falls in frail individuals, especially when programs consist of multiple components [32]. A study by Veronese et al. showed that individuals aged more than 65 years and with sarcopenia had a 1.85-fold greater likelihood of fall-related injury [33]; therefore, the inclusion of one of the most important components—strength training—in a rehabilitation program could be a highly effective prevention strategy in delaying and mitigating the negative effects of sarcopenia and frailty in both early and late phases [34]. On the other hand, Khadanga et al. found that a combination of aerobic exercise and strength training has been shown to be more effective in improving muscle strength, exercise performance, and cardiorespiratory fitness in frail patients compared to using these components separately [35].

In our research, the multicomponent program included aerobic, sensorimotor, strength, and flexibility training, and this program decreased fall risk by 23.08%. Another systematic analysis presents slightly different statistics; however, the researchers only examined multicomponent program effects. Daniel et al. revealed that in training programs that focused on multicomponent exercises, there was a 70% reduction in the risk of falls, a 54% increase in walking speed, an 80% improvement in balance, and a 70% increase in muscle strength among frail individuals [36]. Our study showed that the multicomponent program improved fall risk outcomes. In contrast, patients who received the combined computer-based interactive cardiac program had a 15% and 22.73% reduction in fall, risk as assessed by the overall stability index, compared to the patients in the multicomponent and control groups receiving conventional rehabilitation. It is worth mentioning that before applying additional intervention programs, it is important to assess elderly patients’ frailty status, as it can be a significant predictor of fall risk [37]. A recent meta-analysis performed in 2023, involving studies where frailty was defined by the usage of fall risk, demonstrated that computer-based interactive training increased lower-limb muscle strength, walking speed, and balance, as well as reduced fall risk. However, when frailty was defined by another frailty assessment model, i.e., Fried frailty phenotype criteria, computer-based interactive training improved only walking speed and balance [15]. To sum up, it is necessary to choose the appropriate assessment tool to evaluate frailty, as this may influence the interpretation of fall risk results [3,15].

Many authors underscore the benefits of exercise interventions in reducing the risk of falling among elderly patients [4,38]; however, in a literature search, there is an ongoing discussion regarding what type, frequency, intensity, and duration of exercise interventions are most effective in the rehabilitation of frail patients after open heart surgery. Burton et al. found that multicomponent training, as well as progressing the intensity over time, can contribute to decreasing fall risk in older people [39]. A meta-analysis carried out in 2021 reported that an integrated intervention incorporating resistance, core, and balance training, with a duration exceeding 32 weeks and a frequency of 5 times a week, was effective in reducing the risk of falling [38]. Even though the duration of multicomponent training was shorter in our study (applied three times a week for three weeks), it also resulted in significant changes in fall risk. Another review article showed that multicomponent training is associated with reduced fall risk, as well as improved task performance and memory. Exercise intensity ranged from low to moderate to high, and the duration varied from 5 days to 6, 12, 16, and even 20 weeks [30]. This shows that a long duration does not always lead to a significant difference in the risk of falling. On the one hand, the fairly short time frame for the application of additional exercise interventions in our study allowed us to document significant changes in fall risk. On the other hand, we did not have any possibility to carry out this study on a long-term basis.

A meta-analysis by Schoberer et al. found that computer-based and balance training components had a positive effect on patients’ fall risk, and exercises performed for more than 6 months were more effective than short-term ones [40]. They noted surprising findings that additional long-term interventions causing physical overload could have a negative impact on fall risk, especially among frail residents [40]. Theou et al. revealed that long-term multicomponent training with shorter-duration sessions (30–45 min) might be a better option for this population [41]. These studies provide evidence that when applying additional intervention programs for frail people, it is essential to evaluate an overall duration depending on the type of intervention aimed at maximally improve patients’ physical outcomes. A narrative review published in 2022 reported that when applying computer-based exercises in the rehabilitation of frail adults older than 65 years, they were proven to successfully complement a conventional rehabilitation program, improve motor and cognitive functions, reduce fall risk, and possibly reverse frailty to pre-frailty [17]. A systematic review conducted in 2020 concluded that before making any substantial recommendations on the inclusion of computer-based interventions in the rehabilitation of older adults, it is necessary to pay attention to several factors, especially those that are associated with frailty [42].

All the above research agrees with our study, showing that the application of multicomponent and computer-based interactive interventions in the rehabilitation of patients affected by frailty can improve health outcomes. Even though, in our study, fall risk, as assessed by the overall stability index, was significantly reduced in all groups, when evaluating the obtained data, an appropriate method for frailty assessment and the type of training according to the duration of rehabilitation should be taken into consideration to accurately interpret the results on fall risk. In summary, all these studies and reviews show that additional interventions for frail patients after open heart surgery are effective in reducing fall risk among elderly people, but incorporating additional interventions into a rehabilitation program requires an assessment of the patient’s frailty.

As for gait parameters, in our study, the greatest proportion of significant changes was documented when performing within-group comparisons, and between-group comparisons revealed statistically significant differences mostly between the group receiving multicomponent training and the group that completed the combined computer-based interactive cardiac program. Ruiz-Ruiz et al. in 2021 reported that the most important parameters to be considered in the assessment of frail patients’ gait were double support time, gait speed, stride time, step time, the number of steps per day, or walking percentage per day [43]. When comparing the findings of this study with the results of our study, all groups after rehabilitation in our study showed significant differences in double stance phase, gait speed, stride time, and step time parameters; however, the number of steps per day and walking percentage per day were not evaluated in our study. After the between-group comparison of changes in these gait parameters, we noticed that computer-based interactive training was most effective in improving double stance phase, stride time, and step time parameters as compared with other rehabilitation tools. One of the review articles conducted in 2023 indicated that most studies used various exercise games for lower-limb muscle strength and balance, durations varied from 3 to 15 weeks, and sessions were between 20 and 90 min [15]. Our study showed that the computer-based interactive cardiac program improved fall risk and gait parameters significantly when applied for 20 days, three times a week, with a training session duration of up to 60 min.

Available evidence shows that patients affected by frailty generally have reduced gait speed [44]. The research carried out in 2022 showed that slow gait speed is associated with poorer outcomes after cardiac surgery [45]. In the literature, most of the information is found to be focused on the increase in gait speed after multicomponent training interventions [31,46,47,48]. Simple gait speed measurements are often used in research [49,50]; however, the gold standard for gait analysis is computer-based systems using motion capture techniques and force platforms [51]. The results of our previous study, also including three independent groups, conducted in 2021, indicated that the intervention group receiving training with mechanical devices along with the conventional rehabilitation program showed the greatest improvement in gait speed, but the difference was not statistically significant compared to the other two groups [50]. In this current study, gait speed was also most increased in the computer-based interactive exercise intervention group, but statistically significant difference was not achieved compared to the other groups. A comment by Kim and Oh has emphasized that for accurate measurement of gait speed, it is important to consider not only the timing method (manual or automatic), but also the starting approach (standing or moving start) [52]. In our study, we employed a moving start approach as a protocol for the initiation of gait speed measurements; however, in such measurement setting, the equipment is located against the walking direction, and this might interfere with regular walking. Moreover, the walking path must be about 10 m in length [52]. All these aspects of gait speed measurements, the chosen timing, and starting approach could have influenced and can affect determining a significant difference in gait speed. Thus, further research needs to be conducted.

Some limitations of this study must be addressed. The study was carried out in a single center; therefore, the results cannot be generalized to a broader population. The limited timeframe of the study may not show the long-term impact of the interventions. There was a relatively small sample size and a high patient attrition rate. As vulnerable and mildly frail patients accounted for the largest part of this study population, a question arises as to whether the conclusions can also be applied to severely frail patients to the same extent. The inclusion criteria, such as the capability to walk more than 150 m based on the 6 min walk test and open heart surgery, narrow the possible broader recruitment of participants, as no frail patients who walked less than 150 m or underwent a stenting procedure were enrolled in our study. Kinesiophobia experienced after surgery or lack of skills in walking on the treadmill could have an impact on computerized gait assessment, which could distort the real results.

## 5. Conclusions

Our study showed that all applied interventions were effective in reducing fall risk based on the overall stability index. There were significant differences in the SPPB score only within the groups, while between-group comparisons showed no significant differences. Interactive exercise interventions decreased fall risk based on the overall stability index by one-fifth as compared to the conventional exercise program. Incorporation of interactive exercise interventions in rehabilitation resulted in improved double stance phase, stride time, and step time parameters, but did not have any effect on gait speed as compared to other exercise interventions.

## Figures and Tables

**Figure 1 medicina-61-00206-f001:**
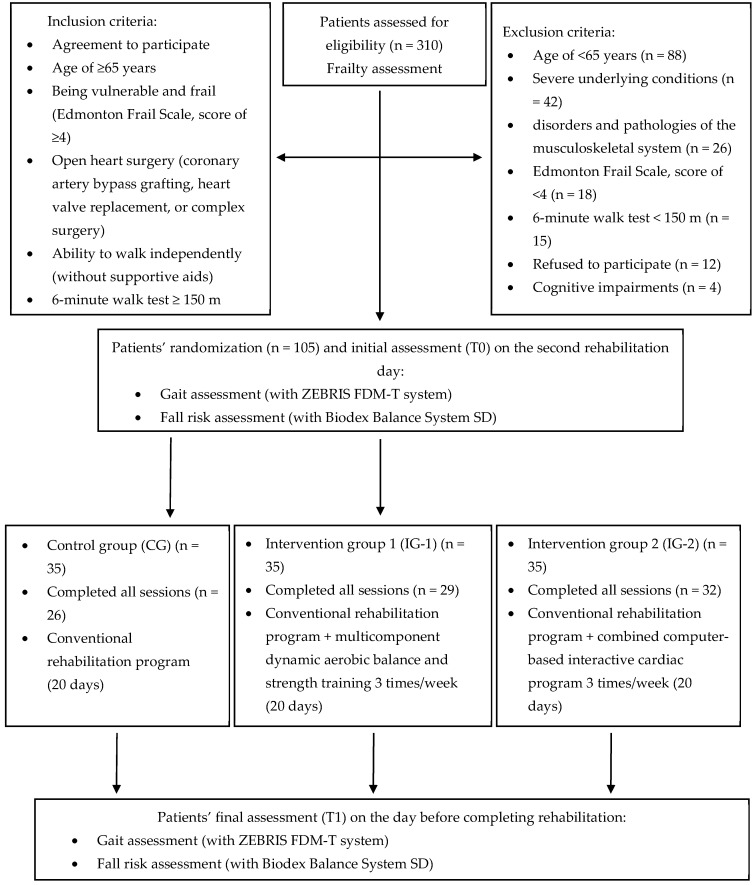
Flowchart of the study.

**Table 1 medicina-61-00206-t001:** Detailed description of the interventions given to the study groups.

Type of Activity	Characteristics	Description
(A) Conventional rehabilitation program		
Aerobic endurance training with a cycle ergometer	Frequency: six times/week Intensity: light-moderate (Borg 9–13) Duration: up to 40–45 min Type: aerobic endurance	Warm-up phase: <50% target intensity 2 min, gradually increasing load by 1–10 W/min up to target intensity within 5–10 min. Exercise phase: workload was started at a light intensity, 37–45% VO_2max_, peak heart rate (HR_peak_) 57–63%, and heart rate reserve from 30–39%, with an initial duration of 5 min, gradually increasing the duration up to 30 min and the intensity to moderate. Recovery phase: gradual reduction of the load within 5 min.
Stretching exercises	Frequency: seven times/week Intensity: light-to-moderate (Borg 9–13) Duration: 10 min Type: passive stretching and ROM	Exercises were performed for major muscle groups after aerobic activities. A static stretch was held for 30 s followed by relaxation up to 30 s.
Breathing exercises	Frequency: seven times/week Intensity: light (Borg 9–11) Duration: 10 min Type: diaphragmatic breathing	Diaphragmatic, supervised exercises combined with hand movements for each patient.
(B) Multicomponent dynamic training		
Aerobic activity	Frequency: three times/week Intensity: moderate (Borg 12–13) Duration: 20–30 min Type: aerobic endurance	Stair climbing, walking, cycling, or other cyclic aerobic activity.
Sensorimotor training	Frequency: three times/week Intensity: light-to-moderate (Borg 9–13) Duration: 15–20 min Type: sensorimotor, balance, coordination	Included postural control, dynamic balance, and coordination. At the beginning, exercises were performed by decreasing the surface of a support base, and later, by using unstable surfaces.
Muscle strength	Frequency: three times/week Intensity: starting from light (30% of 1RM) and increasing to moderate-vigorous (Borg 9–17) Extent: 8–10 exercises (3 sessions, 10 repetitions) Type: strength	Training with elastic resistance bands to increase muscle strength (a band of a particular color based on the resistance level was chosen for each patient individually) and with weights to exercise the major muscle groups of the upper and lower extremities (gradually increasing the weight from 0.5 to 2 kg).
Flexibility	Frequency: three times/week Intensity: light (Borg 9–11) Duration: 5–10 min Type: ROM, stretching	Stretching exercises for the major muscle groups of the upper and lower extremities to increase the range of motion (ROM).
(C) Combined computer-based interactive cardiac program		
Gait improvement and aerobic activity	Frequency: three times/week Intensity: moderate (Borg 12–13) Duration: 20–30 min Type: gait training, aerobic endurance	For gait improvement, we used training with a Biodex GaitTrainer™^3^ treadmill (Shirley, NY, USA) 10–15 min with audio and visual feedback (the walking speed was selected individually according to the T0 test data; range 0.5–3.3 km/h). For gait, endurance, and reaction time improvements, we used the Zebris FDMT training platform (1.9.162; Isny, Germany) for 10–15 min. This platform allows for the virtual creation of obstacles thus simulating the real environment for a patient and at the same time providing them with cognitive tasks (the walking speed was selected individually according to the T0 test data; range 0.5–3.3 km/h). These devices complement each other, and the Zebris FDMT platform further improves patients’ cognitive function.
Sensorimotor training	Frequency: three times/week Intensity: light-to-moderate (Borg 9–13) Duration: ~20 min, each training mode lasted for 3 min Type: sensorimotor, balance, coordination	For sensorimotor, balance, coordination improvements we used a computerized system Biodex Balance SD (v3.06; Shirley, NY, USA) with an integrated platform monitoring the changes in the body’s center of gravity. We included the following programs: postural stability, maze control, limits of stability, random control, weight shift, percent weight-bearing, catch game-based program that improves reaction speed.
Muscle strength	Frequency: three times/week Intensity: starting from light (30% of 1RM) and increasing to moderate-vigorous (Borg 9–17) Extent: 8–10 exercises (3 sessions, 10 repetitions) Type: strength	For muscle strength training, a smart pneumatic technology-based HUR training device was used. During the program involving the major muscle groups of the upper and lower extremities.

**Table 2 medicina-61-00206-t002:** General characteristics of the study population at baseline.

Characteristic	CG	IG-1	IG-2	*p* *
Age, years, median (range)	71 (65; 88)	71 (65; 87)	71 (65; 81)	0.808
Sex, n (%)				
Male	26 (74)	18 (51)	24 (69)	0.114
Female	9 (26)	17 (49)	11 (31)	
Height, m, mean ± SD	1.68 ± 0.08	1.65 ± 0.08 ^a^	1.70 ± 0.08 ^a^	**0.042**
Weight, kg, mean ± SD	80.35 ± 15.94	72.36 ± 14.09 ^a^	82.30 ± 13.52 ^a^	**0.016**
BMI, kg/m^2^, mean ± SD	28.22 ± 4.27	26.51 ± 4.84	28.60 ± 4.71	0.151
LVEF, %, median (range)	50 (30; 55)	50 (30; 58)	50 (34; 60)	0.482
Surgery type, n (%)				
Coronary artery bypass graft surgery	20 (57)	19 (54)	21 (60)	0.901
Heart valve surgery	8 (23)	6 (17)	6 (17)	
AVR-CABG surgery	7 (20)	10 (29)	8 (23)	
EFS, score, median (range)	6 (4; 10)	5 (4; 9)	6 (4; 11)	0.843

BMI, body mass index; LVEF, left ventricular ejection fraction; AVR-CABG, aortic valve replacement in combination with coronary artery bypass grafting; EFS, Edmonton Frail Scale. * Categorical data were compared with the chi-square test; independent normally distributed data were compared with ANOVA; dependent nonnormally distributed data were compared with the Kruskal–Wallis test. Values in bold indicate statistical significance. ^a^ indicates a statistically significant difference between the labeled groups.

**Table 3 medicina-61-00206-t003:** Between-group and within-group comparisons of the overall stability index and Short Physical Performance Battery scores.

Variable	Time Point	CG	IG-1	IG-2	*p* *
Fall risk, score	T0 T1	1.30 (0.60; 2.50) 1.10 (0.60; 2.40) ^a^	1.30 (0.80; 4.30) 1.00 (0.70; 2.10) ^b^	1.15 (0.6; 2.5) 0.85 (0.6; 1.3) ^a,b^	0.330 **<0.001**
***p*** ******		**0.001**	**0.019**	**<0.001**	
SPPB, score	T0 T1	10 (5; 12) 11 (9; 12)	10 (7; 12) 11 (7; 12)	10 (6; 12) 11 (8; 12)	0.090 0.260
***p*** ******		**<0.001**	**<0.001**	**<0.001**	

Values are expressed as median (range). SPPB, Short Physical Performance Battery. * *p* value using Kruskal–Wallis test. ** *p* value using Wilcoxon test. Values in bold indicate statistical significance. ^ab^ indicates a statistically significant difference between the groups labeled with the same letter.

**Table 4 medicina-61-00206-t004:** Between-group and within-group comparisons of gait parameters.

Gait Parameter	Time Point	CG	IG-1	IG-2	*p* *
Geometry					
Foot rotation, degree	T0 T1	12.36 (−0.65; 18.05) ^a^ 9 (2.45; 21.40)	6.80 (−4.40; 15.15) ^ab^ 6.90 (−5.20; 16.75)	9.75 (2.75; 20) ^b^ 9 (2.45; 21.40)	**0.003** 0.096
*p* **		0.273	0.353	**0.048**	
Step length, cm	T0 T1	28.25 (12; 47) ^a^ 32.25 (17.5; 54.50) ^a^	19 (9; 41) ^ab^ 22 (10; 42) ^ab^	35 (17.5; 56) ^b^ 39 (23.50; 57) ^b^	**0.001** **0.001**
*p* **		**0.004**	**0.004**	**0.001**	
Stride length, cm	T0 T1	56 (23; 95) ^a^ 64.50 (35; 110) ^a^	40 (18; 83) ^ab^ 44 (20; 84) ^ab^	69.50 (35; 112) ^b^ 78 (46; 114) ^b^	**0.001** **0.001**
*p* **		**0.006**	**0.004**	**0.001**	
Step width, cm	T0 T1	8.5 (1; 20) ^a^ 9 (2; 20) ^a^	13 (8; 23) ^ab^ 12 (7; 23) ^ab^	11 (4; 19) ^b^ 9 (3; 17) ^b^	**0.001** **0.001**
*p* **		0.549	0.162	0.072	
Phase					
Stance phase, %	T0 T1	68.50 (64.10; 80.50) 66.85 (62.15; 75.15)	70.85 (64.90; 80) 68.28 (64.15; 79.65)	68.35 (63.70; 79.10) 67.10 (61.70; 74.15)	0.098 0.063
*p* **		**0.001**	**0.016**	**0.001**	
Load response, %	T0 T1	18.50 (13.95; 30.40) 16.83 (12.15; 25.15)	20.75 (14.95; 30.15) 18.30 (14.10; 29.70) ^a^	18.05 (13.75; 29.15) 16.83 (11.70; 24.10) ^a^	0.096 **0.039**
*p* **		**0.001**	**0.023**	**0.001**	
Single limb support, %	T0 T1	31.50 (19.65; 36.10) 33.20 (24.80; 37.85)	29.30 (20.15; 35.10) 31.75 (20.35; 35.90) ^a^	32.03 (20.90; 36.25) 33.25 (25.85; 38.20) ^a^	0.086 **0.045**
*p* **		**0.002**	**0.016**	**0.001**	
Pre-swing, %	T0 T1	18.53 (14; 30.30) 16.80 (12.15; 25.15)	20.75 (14.90; 29.90) 18.30 (14.10; 29.70) ^a^	18.03 (13.70; 29.05) 16.78 (11.75; 25.15) ^a^	0.099 **0.039**
*p* **		**0.001**	**0.016**	**0.001**	
Swing phase, %	T0 T1	31.50 (19.50; 35.90) 33.15 (24.85; 37.85)	29.25 (20; 35.10) 31.65 (20.35; 35.85) ^a^	31.90 (20.90; 36.30) 33.18 (25.85; 38.30) ^a^	0.098 **0.039**
*p* **		**0.001**	**0.023**	**0.001**	
Double stance phase, %	T0 T1	37.00 (27.90; 60.80) 33.65 (24.30; 50.30)	41.60 (29.80; 59.80) 36.60 (28.20; 59.30) ^a^	36.10 (27.50; 58.20) 33.55 (23.50; 48.30) ^a^	0.094 **0.037**
*p* **		**0.001**	**0.001**	**0.001**	
Timing					
Step time, s	T0 T1	0.83 (0.49; 1.41) ^a^ 0.71 (0.42; 1.07) ^a^	0.64 (0.49; 1.04) ^ab^ 0.59 (0.44; 0.80) ^ab^	0.89 (0.59; 1.18) ^b^ 0.73 (0.55; 1.10) ^b^	**0.001** **0.001**
*p* **		**0.001**	0.095	**0.001**	
Stride time, s	T0 T1	1.66 (0.97; 2.81) ^a^ 1.41 (0.84; 2.14) ^a^	1.28 (0.97; 2.07) ^ab^ 1.18 (0.89; 1.61) ^ab^	1.77 (1.17; 2.35) ^b^ 1.46 (1.10; 2.18) ^b^	**0.001** **0.001**
*p* **		**0.001**	0.095	**0.001**	
Cadence, steps/min	T0 T1	72.50 (43.00; 124.00) ^a^ 85.00 (56.00; 143.00) ^a^	98.00 (58.00; 124.00) ^ab^ 102.00 (75.00; 139.00) ^ab^	67.50 (52.00; 103.00) ^b^ 82.50 (55.00; 109.00) ^b^	**0.001** **0.001**
*p* **		**0.001**	0.095	**0.001**	
Gait speed, km/h	T0 T1	1.20 (0.5; 2.10) 1.70 (0.80; 3.00)	0.90 (0.50; 1.70) ^a^ 1.30 (0.60; 3.30) ^a^	1.40 (0.80; 2.50) ^a^ 2.00 (0.90; 2.90) ^a^	**0.002** **0.001**
*p* **		**0.001**	**0.001**	**0.001**	

Values are expressed as median (range). * *p* value using Kruskal–Wallis test. ** *p* value using Wilcoxon test. Values in bold indicate statistical significance. ^ab^ indicates a statistically significant difference between the groups labeled with the same letter.

**Table 5 medicina-61-00206-t005:** Comparison of the differences (T1–T0) in gait parameters that significantly differed during T0 assessment among all three independent groups to eliminate the lack of homogeneity.

Gait Parameter	CG T1–T0	IG-1 T1–T0	IG-2 T1–T0	*p* *
Foot rotation, degree	−0.75 (−4.20; 5.95)	0.03 (−3.85; 4.30)	−0.93 (−5.25; 3.20)	0.628
Step length, cm	2 (−8; 19)	5.25 (−9; 18.5)	−3 (−36; 21,5)	0.069
Stride length, cm	4 (−16; 38)	10.5 (−18; 38)	7 (−87; 27)	0.805
bStep width, cm	−1 (−7; 6)	1 (−6; 7)	−1 (−11; 5)	0.117
Step time, s	−0.05 (−0.33; 0.23) ^ab^	−0.14 (−0.49; 0.03) ^a^	−0.13 (−0.52; 0.09) ^b^	**0.030**
Stride time, s	−0.10 (−0.65; 0.47) ^ab^	−0.29 (−0.98; 0.06) ^a^	−0.28 (−1.72; 0.16) ^b^	**0.012**
Cadence, steps/min	11 (−31; 36)	12.50 (−3; 44)	12 (−84; 42)	0.730
Gait speed, km/h	0.40 (−0.10; 1.70)	0.40 (−0.10; 1.30)	0.40 (−1.80; 1.30)	0.486

Values are expressed as median (range). * *p* value using Kruskal–Wallis test. Values in bold indicate statistical significance. ^ab^ indicates a statistically significant difference between the groups labeled with the same letter.

## Data Availability

The data presented in this study are available on request from the corresponding author. The data are not publicly available due to ethical restrictions and data protection policies.

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
