# Peer review of "Effects of Different Exercise Interventions on Fall Risk and Gait Parameters in Frail Patients After Open Heart Surgery: A Pilot Study"

_medicina, 2025, doi:10.3390/medicina61020206_

Round 1
Reviewer 1 Report
Comments and Suggestions for Authors
This manuscript covers the impact of several physical therapy modalities on fall risk and gait parameters in frail postoperative patients undergoing open-heart surgery. This is extremely relevant in the context of today's aging population plagued with high rates of cardiovascular disease among the elderly population. Specifically, this is of great relevance; the research topic of the study also focuses on investigating innovative, interactive interventions for any potential advantages in improving clinical outcomes related to rehabilitation settings. However, it also provides valuable information for rehabilitation strategies for frail patients. Comments on the authors' considerations are outlined below.
Introduction:
• While the literature review is comprehensive, the addition of recent studies on interactive rehabilitation technologies would make the work complete.
Methods
• The research design is appropriate, and the methods are described in sufficient detail to permit replication.
• Randomization and inclusion/exclusion criteria are well defined in addition to the validity of the study.
• The description of interactive interventions is complete, although some technical details could be expanded, such as parameters for progression (e.g., Biodex GaitTrainer™3).
• Ethical guidelines were followed and appropriate approval was obtained.
Results
• The analysis could benefit from a more granular discussion of subgroup differences (e.g., variations based on frailty severity).
Discussion
• The discussion places the findings in the context of existing literature, pointing out the novelty of the contributions of interactive interventions. Certain claims such as the effectiveness of specific interventions can be better supported by additional references or data.
Major Flaws:
1. Sample Size and Generalizability: The limited sample size used and the single-center design mean that any finding may not be generalizable to large populations.
2. Duration of Intervention: The limited timeframe of the study may not show the long-term impact of the intervention on fall risk and gait parameters.
Minor Flaws
1. References: The introduction and discussion may be further enhanced by referring to recent studies on interactive technologies in rehabilitation.
Author Response
|
Comments 1: Introduction: While the literature review is comprehensive, the addition of recent studies on interactive rehabilitation technologies would make the work complete.
|
|
Response 1: Thank you for your comment. We revised the whole introduction. |
|
Comments 2: Methods: The research design is appropriate, and the methods are described in sufficient detail to permit replication. Randomization and inclusion/exclusion criteria are well defined in addition to the validity of the study. The description of interactive interventions is complete, although some technical details could be expanded, such as parameters for progression (e.g., Biodex GaitTrainer™3). Ethical guidelines were followed and appropriate approval was obtained. Response 2: Thank you for your comment. Following the comments by another reviewer also, we revised the Materials and methods section, and all missing information was added to Table 1. “For gait improvements we used trainings with Biodex GaitTrainer™3 treadmill 10–15 min with audio and visual feedback (the walking speed was selected individually according to the T0 test data; range 0.5–3.3 km/h)’’. “For gait, endurance, reaction time improvements we used Zebris FDMT training platform for 10-15 min that allows a virtual creation of obstacles, thus simulating the real environment for a patient and at the same time performing cognitive tasks (the walking speed was selected individually according to the T0 test data; range 0.5–3.3 km/h)”. These devices complement each other, Zebris FDMT additionally improve cognitive patient’s functions.
Comments 3: Results: The analysis could benefit from a more granular discussion of subgroup differences (e.g., variations based on frailty severity). Response 3: Thank you for your comment. We discussed this suggestion with our statistician, and he recommended the following statistical calculations: using the chi-square test to check whether the patients’ numbers by the different subgroups of frailty were equally distributed among all three groups. After performing statistical calculations, there was no significant difference in the distribution of patients by different frailty levels among the groups (p = 0.802). Therefore, presenting additional insignificant data would clutter the Results section, and we would like to avoid this.
Comments 4: Discussion: The discussion places the findings in the context of existing literature, pointing out the novelty of the contributions of interactive interventions. Certain claims such as the effectiveness of specific interventions can be better supported by additional references or data. |
|
Response 4: Thank you for your comment. We revised the whole discussion.
Comments 5: Sample Size and Generalizability: The limited sample size used and the single-center design mean that any finding may not be generalizable to large populations. Response 5: Thank you for your comment. We mentioned this aspect as one of the limitations of our study: ”The study was carried out in a single center; therefore, the results cannot be applicable for a broader population”.
Comments 6: Duration of Intervention: The limited timeframe of the study may not show the long-term impact of the intervention on fall risk and gait parameters. Response 6: Thank you for your comment. We corrected this information in the paragraph of study limitations: “Short duration of rehabilitation” we changed to “The limited timeframe of the study may not show the long-term impact of the interventions”.
Comments 7: References: The introduction and discussion may be further enhanced by referring to recent studies on interactive technologies in rehabilitation. Response 7: Thank you for your comment. We searched the available literature and cited new references in the manuscript.
|

Reviewer 2 Report
Comments and Suggestions for Authors
I am grateful to the editor for the opportunity to review the manuscript by Vitalija Stonkuvienė et al. "Effects of different exercise interventions on fall risk and gait parameters in frail patients after open heart surgery: a pilot study". In this article, the authors studied the effectiveness of three different exercise training programs in patients with frailty after cardiac surgery. The training programs were studied during a 20-day treatment in a rehabilitation center, which allowed the authors to obtain new scientific facts.
While reviewing, I had the following comments and questions:
1. In terms of a number of Gait testing parameters, the IG-1 group differs significantly from the other two groups (Table 3). According to the baseline parameters (Table 1), the groups differed significantly in weight and height. These differences may have been due to the predominance of women in this group. At the time of the baseline examination, the differences between the groups were insignificant. However, 18 patients in the groups did not complete the study, and there is no data on the comparability of the groups at the time of study completion. It is possible that the observed differences in the training results are due to gender differences (predominance of women in the IG-1 group). Therefore, additional data are needed to exclude this assumption.
2. The high frequency of patient dropouts from the study requires providing data on the reasons for dropping out of the study in individual groups (i.e., is there an effect of the training program on the frequency and reasons for dropping out of the study?).
3. The Discussion section requires correction. It would be appropriate to begin it with the main result obtained in the study (i.e., with the second paragraph). In this case, the first paragraph of the Discussion section is redundant.
Author Response
|
Comments 1: In terms of a number of Gait testing parameters, the IG-1 group differs significantly from the other two groups (Table 3). According to the baseline parameters (Table 1), the groups differed significantly in weight and height. These differences may have been due to the predominance of women in this group. At the time of the baseline examination, the differences between the groups were insignificant. However, 18 patients in the groups did not complete the study, and there is no data on the comparability of the groups at the time of study completion. It is possible that the observed differences in the training results are due to gender differences (predominance of women in the IG-1 group). Therefore, additional data are needed to exclude this assumption. |
|||||||||||||||||||||||||
|
Response 1: Thank you for your comment. In order to rule out the possible effect of the female gender after drop-out, we performed an additional analysis with the SPSS program. Please find the contingency and statistics tables below.
As one can see, after the drop-out, there was no significant difference in the distribution of patients by gender among the groups as well. Therefore, we can rule out the effect of gender. As all similar studies in this area usually provide the comparison of demographic and anthropometric patients’ characteristics at baseline, we would like to keep our initial table.
|
|||||||||||||||||||||||||
|
Comments 2: The high frequency of patient dropouts from the study requires providing data on the reasons for dropping out of the study in individual groups (i.e., is there an effect of the training program on the frequency and reasons for dropping out of the study?). Response 2: Thank you for your comment. First, the year 2021 was the second year of the COVID pandemic, and in general the pandemic was very unfavorable for conducting any type of research. Second, the population of our study comprised of elderly frail patients who underwent open heart surgery. This population was especially sensitive and prone to diseases, including COVID infection. The table below shows the reasons for dropping out of the study by the individual groups Table. Reasons for dropping out of the study by the individual groups.
As one can see, the greatest number of patients who dropped out was from the control group that received only the conventional rehabilitation program; therefore, the intensity of the exercises as the important reason for dropping out of the study can be ruled out. As we did not want to include an additional table (the content of which is quite negligible) in our manuscript, we amended the text by providing the exact numbers of patients who dropped out for each group. Now this sentence reads as follows: ‘’The main reasons for withdrawal from the study were as follows: worsened health status (n = 11; 4, 4 and 3 patients in the CG, IG-1, and IG-2, respectively), left home earlier (n = 5; 3 and 2 patients in the CG and IG-1, respectively), and declined to continue participation in the study (n = 2; both patients from the CG)”.
Comments 3: The Discussion section requires correction. It would be appropriate to begin it with the main result obtained in the study (i.e., with the second paragraph). In this case, the first paragraph of the Discussion section is redundant. Response 3: Thank you for your comment. However, we are not sure if we understood it correctly. The previous version of our manuscript had the discussion starting with the main results obtained in the study, and they were presented in one paragraph. Therefore, mentioning the first and second paragraphs creates uncertainty. As we amended the information in the discussion in general, we hope that our amendments will be relevant.
|

Reviewer 3 Report
Comments and Suggestions for Authors
Introduction
Although the title of the study indicates that the focus of the study is open heart surgery, fragile patients, fall risk and walking parameters, it is seen that these elements are not emphasized enough in the introduction and the effects of exercise interventions are not clearly explained. In addition, there is no explanation about the "exercise interventions" and "effects" mentioned in the title.
The originality and contribution of the study are not stated. The general health and cardiovascular disease data in the introduction constitute a mass of information that does not serve the purpose of the study and a more specific context should be given. This information cannot be directly linked to the main purpose of the study, namely to examine the effect of exercise interventions on fall risk and walking parameters.
As a result, the purpose and focus of the study should be clearly stated, the effects of exercise interventions should be focused on and unnecessary details should be avoided. The introduction should be revised to reflect the originality and purpose of the study.
The research hypothesis should be given.
Line 95-97: "The aim of this study was to evaluate and compare the effects of different physical therapy interventions on fall risk and gait parameters of frail patients after open heart surgery during rehabilitation."
The term "exercise interventions" was used in the research title. The terms "physical therapy interventions" and "exercise interventions" have different meanings. Using "physical therapy interventions" in the purpose statement while using "exercise interventions" in the title creates a terminological inconsistency. If the focus of the study is to examine the effects of exercise interventions, it would be more appropriate to correct the purpose statement to "exercise interventions" to be consistent with the title. Otherwise, it may lead to confusion about the scope of the study.
Line 43-60: As understood from the research title and purpose, the focus of the study is the effect of exercise interventions on the fall risk and gait parameters parameters in thin patients after open heart surgery. In this context, cardiovascular diseases and general aging statistics are not considered to be directly related to the subject of this study. In particular, it would be more effective to link to literature that more directly serves the aims of the study and to data on frail patients after open heart surgery. It would be more effective to emphasize more specialized and directly relevant information on the subject of the study (rehabilitation of frail patients after open heart surgery, fall risk, and exercise interventions).
Line 77-94: When looking at this paragraph, the information given first talks about the physiological difficulties faced by frail patients after open heart surgery and important issues such as the risk of falling, but then it moves on to modern rehabilitation techniques (such as pressure platforms, video cameras). It seems that this transition is a bit choppy and the flow of the topic is scattered. furthermore, the relationship of these tools to exercise interventions is not clearly stated.
Line 86-89: The application of multicomponent programs and interactive tools in rehabilitation of these patients is limited, and research on this topic is scarce.
While the purpose of the study was to examine the effects of exercise interventions on frail patients undergoing open heart surgery, the information provided here addresses the increasing popularity of interactive computerized programs for balance and gait training with the use of traditional tools. This shift may make it more difficult for the reader to follow the purpose of the study because the subject suddenly shifts to computerized systems and deviates from the purpose of the study.
Furthermore, the statement "multi-component programs and interactive tools have limited application" does not include a reference to the literature on which it is based. In the context of a scientific study, it is important to cite the sources on which such claims are based.
2. Materials and Methods
Under the Study Assessment heading (lines 118-135), information about the design of the study and the participants is provided. The heading 'Study design and participants' does not contain information about the research design. The method section should clearly define the structural and methodological aspects of the study. Titles in the method section should be updated. For example, the participants heading should include: Basic demographic information such as age, gender, and health status of the participants, Inclusion and exclusion criteria, Power analysis used to determine the number of participants, Total number of participants participating in the study, Gender distribution of participants (e.g., ratios of women and men).
2.1. Study design and participants
The age range of the participants included in the study should be specified. Instead of stating "65 years and older", specifying a specific age range (e.g. 65-85 years) provides more meaningful and detailed information, as different age ranges may differ in terms of physical health status and rehabilitation needs.
The research design should be presented in a clearer and more sequential manner, allowing the reader to understand the study process and application steps more easily. The research design should be presented in a more understandable and sequential manner. Which application was applied to which participant and which tests were applied should be stated more clearly.
2.2. Study Assessment
Line 122: 105 were eligible for this clinical study
No information was given as to whether a power analysis was performed in determining the research group.
2.3. Study interventions
A decision should be made to use either the term exercise or physical therapy consistently throughout the study. This dilemma should be clarified according to the purpose and context of the study. If the focus of the study is to examine the effects of specific types of exercise, the term "exercise" may be more accurate.
This protocol seems quite dense and complicated in terms of implementation details. Information such as the sequence of training programs, durations, intensities and equipment used should be presented in a clearer and more understandable way. In addition, more explanation should be provided on how the intensity of the exercises is determined by the scores given on the Borg scale and how this system is applied to each participant.
In particular, in addition to the names of the technologies and devices used, details such as the duration of the exercise, the intensity, and how the device used works for each type of exercise need to be specified more clearly. For example, it would be useful to explain in more detail how devices such as the “Biodex GaitTrainer™3 treadmill” and the “Zebris FDM-T” interact and the purpose of using these devices. In addition, more information may need to be provided on how the intensity and duration of the exercises are determined. It is important to define the intensity determined for each type of exercise more clearly with the scores given on the Borg scale.
Intervention protocols are written in a very complicated way. The steps of the exercise and rehabilitation process specified in the protocol should be presented in a clear and orderly manner. The number of minutes and intensity of interventions should be explained step by step.
Line 176-177: Before starting the program, each participant underwent testing with a cycle ergometer to determine optimal and to ensure individual physical workload.
It is important to provide information about how this type of test is administered, its validity, and its reliability.
Line 175: All participants were provided with a conventional 175 physical therapy program.
If all participants received a physiotherapy program, it should be more clearly stated whether the other three intervention programs were applied additionally.
Line 187: In addi tion, patients performed breathing and stretching exercises for 15–20 min, 7 days a week
The fact that this exercise was given only to the control group creates ambiguity in the research design. It should be clarified why only the control group did these exercises and not the other groups. In addition, it should be stated in more detail how the effectiveness of this exercise was evaluated and on what criteria the results were obtained.
4. Discussion
The discussion section requires a more in-depth comparison of the research findings with the literature. The current discussion does not sufficiently relate the findings of the research to similar studies in the literature. The accuracy and meaning of the results obtained should be more clearly demonstrated by comparing with the existing literature. The parallels and differences between the findings obtained in similar studies in the literature and the results of the research should be emphasized, and the meaning of the findings, the factors that are important and how the results can be generalized should be emphasized.
The main findings of the study are not clearly stated. It is expected that data will be presented on the main results of the study, such as differences between intervention groups, recovery rates, and changes in important parameters.
Line 309: different physical therapy interventions
"exercise interventions" "physical therapy interventions" should be decided
Line 314: physical therapy programs
Line 315-318: Patients who received the combined computer based interactive cardiac program had a 15% and 22.73% reduced fall risk assessed by overall stability index as compared with the patients in the group receiving multicomponent dynamic aerobic balance and strength training and in the control group receiving conventional rehabilitation.
These groups should be clearly defined in the method section. In the method section, there are groups as 1) The control group (CG), 2) The first intervention group (IG-1), 3) Patients randomized to the second intervention group (IG-2). In the method section, it should be stated which intervention groups received which programs and how these groups were formed.
Line 320-330: This information includes a discussion of which findings of the study? Focusing on factors directly related to the content of the study and based only on the findings obtained in the discussion section would be a more scientific and systematic approach. When making a discussion based on the findings of the study, only data and literature related to the results of the study should be evaluated. Authors need to make a more direct discussion about the data they obtained. Literature information not directly related to the findings of the study may damage the clarity and focus of the study. In addition, after mentioning conditions such as osteoporosis, arthritis, and low back pain in this paragraph, moving on to risk factors such as gender creates a situation that disrupts the integrity of the subject and exhibits a scattered narrative.
Line 331-332: is very scarce…Many authors…
There is a contradiction in this paragraph. These two statements contradict each other, because if there is ‘very little’ literature, it would not be correct to use the expression ‘many authors’.
Line 334-336: however, there is an ongoing discussion what type, frequency, inten- 334 sity, and duration of physical therapy interventions are most effective in the rehabilitation 335 of frail patients after open heart surgery…
This should be clearly emphasised in the methodology section of this study. intervention protocols should be addressed comprehensively in terms of frequency, duration and intensity.
During the discussion of the research method, it is important to compare the applied intervention protocols with other research protocols in the existing literature. This comparison is necessary to evaluate the effectiveness of the interventions. Findings on the effectiveness of similar interventions in the literature should be presented and how they overlap or differ from the results of this study should be discussed.
Author Response
|
Comments 1: Introduction: Although the title of the study indicates that the focus of the study is open heart surgery, fragile patients, fall risk and walking parameters, it is seen that these elements are not emphasized enough in the introduction and the effects of exercise interventions are not clearly explained. In addition, there is no explanation about the "exercise interventions" and "effects" mentioned in the title. |
||||||||||||||||||||||||||||||||||||||||||
|
Response 1: Thank you for your comment. We revised the whole introduction.
Exercise-based cardiac rehabilitation (CR) is widely recommended for patients after cardiac events [1]. CR is considered as the standard of care to facilitate the recovery from cardiovascular disease (CVD) using exercise therapy, management of nutrition, modification of risk factors, psychosocial support, and patient education. It is known that functional improvement occurs through rehabilitation exercises, education, and training. Exercise-based CR increases VO2 maximum and endurance, and it contributes to reduced mortality and rehospitalization [2]. Research shows that the benefits of CR can extend beyond improvements in the secondary prevention of CVD to alleviate age-related health deficits [3]. Elderly patients are affected by CVD, and with an aging population, the number of elderly people with CVD will increase dramatically [4;5]. There is a strong bidirectional association between CVD and frailty, and frail older adults are at a greater risk of developing CVD [6]. Most commonly frailty is defined as an age-related biological syndrome characterized by a decreased biological reserve due to impairments in the regulation of multiple physiologic systems [7]. This leads to increased vulnerability of an individual to physiologic stressors, and it is linked to adverse outcomes such as disability, hospitalization, or death [8]. There are several frailty phases ranging from complete absence of frailty to vulnerable state followed by physical frailty. Vulnerable state or prefrailty can increase the risk of developing frailty [9]. According to one meta-analysis, the overall prevalence of frailty ranges from 40% to 50% depending on the frailty tool used [3]. Frailty is closely related to falls, and it is the second leading unintentional cause of injury and death among older adults [10]. Falls, as one of the major “geriatric syndromes”, like the walking process, are the result of neurological, musculoskeletal, nutritional, and cardiovascular integration [11]. Open heart surgery that is frequently performed among patients with frailty can cause physiological stress. Therefore, 28% of elderly people arriving at inpatient rehabilitation already have decreased skeletal muscle strength and physical performance, and this has a huge impact on balance and general performance to carry out daily activities [12], including walking [13]. Exercise-based CR sets a goal to improve fall risk and gait parameters in frail patients after open heart surgery. There are no universally accepted and evaluated standards for exercise-based CR regarding its contents, frequency, time, and intensity for frailty patients after cardiac intervention [1]. Traditionally, conventional training means are used for balance and gait training in frail patients after open heart surgery during rehabilitation. Research indicates the benefits of different interventions such as multicomponent programs [14] or interactive programs [15]. Based on the literature, a multicomponent program should include elements of aerobics, strength, balance, and flexibility. These elements improve gait and reduce the risk of falling [16]. Studies have shown that frailty patients are recommended a combination of aerobics and strength training that increases muscle mass and improves strength, function, and balance. Muscle strength and balance training is a key component of exercise programs to reduce frailty and fall risk scores in older adults [14;16]. In computer-based rehabilitation, pressure platforms and video cameras are used for computerized balance and gait analysis that allow to objectively determine balance and gait parameters as well as to evaluate the effectiveness of the applied program [17]. Interactive, computerized, physical performance-enhancing programs that are safe, improve balance, reduce physical impairment, combine real-time motion detection and feedback to the user and interact with virtual objects and evaluate movement accuracy, speed and activity duration [15]. These interventions improve lower limb muscle strength, walking speed, balance, and fall risk [15]. Computer-based interactive programs are useful, help motivate patients to exercise, and are increasingly being included into CR [18]. The application of multicomponent programs and interactive tools in rehabilitation of frailty patients after open heart surgery is limited [19;15], and research on this topic is scarce. To our knowledge, there are no published articles that evaluated and compared the effects of multicomponent dynamic and combined computer-based interactive cardiac training programs. Therefore, the aim of this study was to evaluate and compare the effects of the conventional rehabilitation program and different complementary exercise programs (multicomponent dynamic and combined computer-based interactive cardiac training programs) on fall risk and gait parameters of frail patients after open heart surgery during rehabilitation. We hypothesized that multicomponent dynamic and combined computer-based interactive cardiac training programs would reduce fall risk and improve gait parameters better than the conventional rehabilitation program.
|
||||||||||||||||||||||||||||||||||||||||||
|
Comments 2: Introduction: The originality and contribution of the study are not stated. The general health and cardiovascular disease data in the introduction constitute a mass of information that does not serve the purpose of the study and a more specific context should be given. This information cannot be directly linked to the main purpose of the study, namely to examine the effect of exercise interventions on fall risk and walking parameters. As a result, the purpose and focus of the study should be clearly stated, the effects of exercise interventions should be focused on and unnecessary details should be avoided. The introduction should be revised to reflect the originality and purpose of the study. Response 2: Thank you for your comment. We revised the whole introduction.
Comments 3: Introduction: The research hypothesis should be given. Response 3: Thank you for your comment. After the aim of our study, we added the research hypothesis. Therefore, the aim of this study was to evaluate and compare the effects of the conventional rehabilitation program and different complementary exercise programs (multicomponent dynamic and combined computer-based interactive cardiac training programs) on fall risk and gait parameters of frail patients after open heart surgery during rehabilitation. We hypothesized that multicomponent dynamic and combined computer-based interactive cardiac training programs would reduce fall risk and improve gait parameters better than the conventional rehabilitation program.
Comments 4: Introduction: Line 95-97: "The aim of this study was to evaluate and compare the effects of different physical therapy interventions on fall risk and gait parameters of frail patients after open heart surgery during rehabilitation." The term "exercise interventions" was used in the research title. The terms "physical therapy interventions" and "exercise interventions" have different meanings. Using "physical therapy interventions" in the purpose statement while using "exercise interventions" in the title creates a terminological inconsistency. If the focus of the study is to examine the effects of exercise interventions, it would be more appropriate to correct the purpose statement to "exercise interventions" to be consistent with the title. Otherwise, it may lead to confusion about the scope of the study. |
||||||||||||||||||||||||||||||||||||||||||
|
Response 4: Thank you for your comment. We agree with this comment. We changed the term "physical therapy interventions" to "exercise interventions". Therefore, it is more accurate and better reflects our study.
Comments 5: Introduction: Line 43-60: As understood from the research title and purpose, the focus of the study is the effect of exercise interventions on the fall risk and gait parameters parameters in thin patients after open heart surgery. In this context, cardiovascular diseases and general aging statistics are not considered to be directly related to the subject of this study. In particular, it would be more effective to link to literature that more directly serves the aims of the study and to data on frail patients after open heart surgery. It would be more effective to emphasize more specialized and directly relevant information on the subject of the study (rehabilitation of frail patients after open heart surgery, fall risk, and exercise interventions). Response 5: Thank you for your comment. We revised the whole introduction.
Comments 6: Introduction: Line 77-94: When looking at this paragraph, the information given first talks about the physiological difficulties faced by frail patients after open heart surgery and important issues such as the risk of falling, but then it moves on to modern rehabilitation techniques (such as pressure platforms, video cameras). It seems that this transition is a bit choppy and the flow of the topic is scattered. furthermore, the relationship of these tools to exercise interventions is not clearly stated. Response 6: Thank you for your comment. We revised the whole introduction.
Comments 7: Introduction: Line 86-89: The application of multicomponent programs and interactive tools in rehabilitation of these patients is limited, and research on this topic is scarce. While the purpose of the study was to examine the effects of exercise interventions on frail patients undergoing open heart surgery, the information provided here addresses the increasing popularity of interactive computerized programs for balance and gait training with the use of traditional tools. This shift may make it more difficult for the reader to follow the purpose of the study because the subject suddenly shifts to computerized systems and deviates from the purpose of the study. Furthermore, the statement "multi-component programs and interactive tools have limited application" does not include a reference to the literature on which it is based. In the context of a scientific study, it is important to cite the sources on which such claims are based. Response 7: Thank you for your comment. We revised the whole introduction.
Comments 8: Materials and Methods: Under the Study Assessment heading (lines 118-135), information about the design of the study and the participants is provided. The heading 'Study design and participants' does not contain information about the research design. The method section should clearly define the structural and methodological aspects of the study. Titles in the method section should be updated. For example, the participants heading should include: Basic demographic information such as age, gender, and health status of the participants, Inclusion and exclusion criteria, Power analysis used to determine the number of participants, Total number of participants participating in the study, Gender distribution of participants (e.g., ratios of women and men). Response 8: Thank you for your comment. While preparing our manuscript, we followed various guidelines for scientific writing in health sciences. All these guidelines (including the STROBE statement) recommend providing the information in Materials and methods adhering to the following aspects: 1. Study design and setting; 2. Patient selection; 3. Patient assignment; 4. Interventions; 5. Measures; 6. Statistical methods.
All the data regarding patients’ demographic, clinical and anthropometric characteristics, according to these guidelines, have to be presented at the beginning of Results. Therefore, we would like to keep the information related to patients’ characteristics as it is, i.e. in the Results section. However, in order to make all the information in Materials and methods more clearly presented, but avoiding the unnecessary clutter and repetition of the information, we reorganized some parts of this section, added additional chapters and improved their titles as well as inserted a new table depicted all the interventions in detail. Now the Materials and methods section appears to be as follows: 2. Materials and Methods 2.1 Study design, setting, and participant selection This prospective randomized experimental study was conducted at Kulautuva Hospital of Rehabilitation, Hospital of Lithuanian University of Health Sciences Kauno Klinikos, from July 2021 to November 2023. Kauno Regional Biomedical Research Ethics Committee approved this study (No. BE-2-83, dated July 22, 2021). The study was carried out in accordance with the ethical principles outlined in the Declaration of Helsinki and was registered in a clinical trial registry at ClinicalTrials.gov (registration No. NCT06385041). All patients who arrived at Kulautuva Hospital of Rehabilitation and who had undergone open heart surgery, i.e. coronary artery bypass grafting, heart valve replacement, or complex surgery, and met the inclusion criteria were invited to participate in the study. Patients were included in the study if they were: 1. Older than 65 years, 2. Vulnerable or frail (a score of ≥ 4 on the Edmonton Frail Scale, EFS), 3. Underwent open heart surgery, 4. Able to walk independently (without supportive aids), 5. Walked ≥ 150 m based on the 6-minute walk test, 6. Signed an informed consent form. The exclusion criteria were as follows: 1. Refusal to participate in the study, 2. Disorders and pathologies of the musculoskeletal system, 3. Cognitive impairment, 4. Severe underlying conditions (mental, visional, and hearing impairments, heart failure, severe anemia, complications of postoperative wound healing), 5. Other acute conditions that could limit active participation in exercise programs.
2.2. Participant assessment and assignment
A total of 310 patients who underwent open heart surgery were admitted and registered at Kulautuva Hospital of Rehabilitation, Hospital of Lithuanian University of Health Sciences Kauno Klinikos. Of them, 205 patients did not meet the inclusion criteria. All patients who arrive to cardiac rehabilitation after open heart surgery are given 20-day inpatient rehabilitation. On the first day of arrival to cardiac rehabilitation, the candidates who met the inclusion criteria were introduced to the course of the study, and they received information about the study both verbally and in writing. After answering all the questions raised as well as reading and signing an informed consent form, each patient received a unique code for data coding. After patients’ examination by a physical medicine and rehabilitation physician, each patient is given a rehabilitation plan. On the second day of rehabilitation, patients were clinically examined and the assessment of frailty status, fall risk, and gait was done. After this assessment, patients were randomly allocated to one of the three groups at a 1:1:1 ratio using the Research Randomizer program. During the entire study period, all patients received therapeutic massage, physical factors, counselling by a psychologist and a social worker, medication treatment, management of risk factors, and protein-enriched diet.
2.3 Study interventions
Patients were allocated to the following three groups: 1. Control group (CG) that underwent only conventional rehabilitation program (Table 1, part A); 2. Intervention group 1 (IG-1) that underwent the conventional rehabilitation program (Table 1, part A) plus multicomponent dynamic training (Table 1, part B); 3. Intervention group 2 (IG-2) that underwent the conventional rehabilitation program (Table 1, part A) plus the combined computer-based interactive cardiac program (Table 1, part C).
Table 1. Detailed description of the interventions given the study groups
Before starting the program, each participant underwent testing with a cycle ergometer to determine optimal and to ensure individual physical workload. The cardiovascular stress test was performed using a cycle ergometer (Ergoline GmbH, Germany). During this test, electrocardiogram (ECG), heart rate, blood pressure, and well-being of the subject were monitored, and all complaints were documented. The load was started with 25 W, and it was gradually increased by 12.5 W every minute until the patient became exhausted or other criteria for study discontinuation were obvious: significant changes in ECG, hypotension (decrease in systolic BP > 20 mmHg), marked hypertension (> 230/120 mmHg), chest pain, pronounced shortness of breath, or other exercise-limiting symptoms. The intensity of the achieved load was evaluated in W and metabolic units (MET). One MET is the amount of oxygen while sitting consumed per kg of body weight in 1 min and equal to 3.5 ml/kg/min [20]. The cardiovascular stress test is the gold standard for the diagnostic evaluation of exercise intolerance especially in cardiac patients and for setting an individualized prescription of structured exercise training program [21;22;23]. This test is typically performed on a treadmill or stationary bike. The test shows very good-to-excellent validity (r = 0.85–0.97) and moderate-to-high reliability (intraclass correlation coefficient = 0.85–0.91) [24].
2.4 Study measures
All the patients were assessed twice: at the beginning of rehabilitation, i.e. on the 2nd rehabilitation day (T0) and on the day before completing rehabilitation (T1). During both assessments, the same data were evaluated and collected, except for the EFS that was applied only during T0 assessment and the Borg Rating of Perceived Exertion (RPE) scale that was used during each exercise session [25]. The EFS was employed to evaluate frailty. This scale consists of 9 domains: cognition, general health status, functional independence, social support, medication use, nutrition, mood, continence, and functional performance. Each domain is assessed on a 0–2-point scale; the total maximal score available is 17. Based on the score obtained, the level of frailty is evaluated as follows: score of 0 to 3, fit; score of 4 to 5, vulnerable; score of 6 to 7, mild frailty; score of 8 to 9, moderate frailty; and score of 10 and more, severe frailty [26]. This biomedical study enrolled only those patients who were vulnerable and mildly-to-severely frail, i.e. patients who got a score of 4 and more. Fall risk was assessed using the computerized Biodex Balance system SD. This system utilizes a multi-axial tilting platform allowing to assess patient’s fall risk and compare the results in a certain age group. During dynamic fall risk assessment, the platform was set to permit up to 20° of deflection in all directions (level 8). Subjects were instructed to maintain an upright posture 3 times for 20 s without falling and grasping the handrails with eyes open and feet shoulder-width apart. The subject applied pressure to the platform, and anterior, posterior, medial, and lateral sways were recorded. Then, a derivative overall stability index ranging from 0.0 (optimal balance / equivalent of low fall risk) to 5.0 (poor balance / equivalent of high fall risk) was generated [27]. Gait analysis was done using a treadmill with an embedded pressure plate Zebris, FDM, GmbH, Germany). During the study, the following 3 groups of parameters were evaluated: geometry (foot rotation, step length, stride length, step width), phase (stance, load response, single limb support, pre-swing, swing, double stance), and timing (step time, stride time, cadence, and gait speed). Patients were asked to walk barefoot at their preferred gait speed. Gait testing lasted for 30 seconds. The entire assessment was captured with two video cameras set up in the frontal and sagittal planes. Gait parameters were recorded in the software [28]. The Borg Rating of Perceived Exertion (RPE) scale is the most widely used subjective tool to measure an individual’s effort and exertion during physical activity. Patients were asked to rate their exertion on a 6–20-point scale during exercises, combining all sensations of physical stress and fatigue. The patient's heart rate, parameters of heart rhythm (extrasystoles), respiratory rate, sweating, and medications used (beta-blockers) were also assessed in parallel. In our study, we used a score of 9–11 corresponding to light physical activity; 12–13, to moderate; 14–17, to vigorous; and 18–20, to very vigorous [25]. The Short Physical Performance Battery (SPPB) was used to evaluate the following three components: balance, gait speed, and timed chair stand. Each component is scored from 0 to 4, and the overall SPPB score ranging from 0 to 12 is obtained by adding the scores of individual components. A score of 0 indicates the worst performance, and a score of 12, the best performance [29].
Comments 9: Study design and participants: The age range of the participants included in the study should be specified. Instead of stating "65 years and older", specifying a specific age range (e.g. 65-85 years) provides more meaningful and detailed information, as different age ranges may differ in terms of physical health status and rehabilitation needs. Response 9: Thank you for your comment. For the inclusion criteria we could not specifically set the upper age limit as all patients older than 65 years were eligible for the study; therefore, there was no upper age limit. Later, in the results section we clearly state the range for the age of the patients who took part in the study.
Comments 10: Study design and participants: The research design should be presented in a clearer and more sequential manner, allowing the reader to understand the study process and application steps more easily. The research design should be presented in a more understandable and sequential manner. Which application was applied to which participant and which tests were applied should be stated more clearly. Response 10: Thank you for your comment. As mentioned above, we revised the Materials and methods section and improved it regarding clarity. We added a new table where it is clearly presented what interventions were given to each group. Moreover, all the information regarding the study flow is summarized in Figure 1.
Comments 11: Study Assessment: Line 122: 105 were eligible for this clinical study. No information was given as to whether a power analysis was performed in determining the research group. Response 11: Thank you for your comment. We added the following information: “The outcome measure SPPB was used to calculate a sample size. A target sample size of 105 (35 per group) was calculated to ensure 80% power (β = 0.2) to detect a minimum significant change of 1 point in the SPPB (α = 0.05).”
Comments 12: Study interventions: A decision should be made to use either the term exercise or physical therapy consistently throughout the study. This dilemma should be clarified according to the purpose and context of the study. If the focus of the study is to examine the effects of specific types of exercise, the term "exercise" may be more accurate. Response 12: Thank you for your comment. As we mentioned in our reply to comment 4, we changed the term "physical therapy interventions" to "exercise interventions" as it is more accurate and better reflects our study.
Comments 13: Study interventions: This protocol seems quite dense and complicated in terms of implementation details. Information such as the sequence of training programs, durations, intensities and equipment used should be presented in a clearer and more understandable way. In addition, more explanation should be provided on how the intensity of the exercises is determined by the scores given on the Borg scale and how this system is applied to each participant. Response 13: Thank you for your comment. As mentioned above, we revised the Materials and methods section and improved it regarding clarity. We added a new table where it is clearly presented what interventions were given to each group. Moreover, we added additional information about the intensity of the exercises that was determined by the scores on the Borg scale. “The Borg Rating of Perceived Exertion (RPE) scale is the most widely used subjective tool to measure an individual’s effort and exertion during physical activity. Patients were asked to rate their exertion on a 6–20-point scale during exercises, combining all sensations of physical stress and fatigue. The patient's heart rate, parameters of heart rhythm (extrasystoles), respiratory rate, sweating, and medications used (beta-blockers) were also assessed in parallel. In our study, we used a score of 9–11 corresponding to light physical activity; 12–13, to moderate; 14–17, to vigorous; and 18–20, to very vigorous [25].”
Comments 14: Study interventions: In particular, in addition to the names of the technologies and devices used, details such as the duration of the exercise, the intensity, and how the device used works for each type of exercise need to be specified more clearly. For example, it would be useful to explain in more detail how devices such as the “Biodex GaitTrainer™3 treadmill” and the “Zebris FDM-T” interact and the purpose of using these devices. In addition, more information may need to be provided on how the intensity and duration of the exercises are determined. It is important to define the intensity determined for each type of exercise more clearly with the scores given on the Borg scale. Response 14: Thank you for your comment. As mentioned above, we revised the Materials and methods section and improved it regarding clarity. We added a new table where it is clearly presented what interventions were given to each group. Moreover, we included new information the Biodex GaitTrainer™3 treadmill and the Zebris FDM-T.
Comments 15: Study interventions: Intervention protocols are written in a very complicated way. The steps of the exercise and rehabilitation process specified in the protocol should be presented in a clear and orderly manner. The number of minutes and intensity of interventions should be explained step by step. Response 15: Thank you for your comment. As mentioned above, we revised the Materials and methods section and improved it regarding clarity. We added a new table where it is clearly presented what interventions were given to each group. Moreover, all the information regarding the study flow is summarized in Figure 1.
Comments 16: Study interventions: Line 176-177: Before starting the program, each participant underwent testing with a cycle ergometer to determine optimal and to ensure individual physical workload. It is important to provide information about how this type of test is administered, its validity, and its reliability. Response 16: Thank you for your comment. We added additional information to the manuscript. “Before starting the program, each participant underwent testing with a cycle ergometer to determine optimal and to ensure individual physical workload. The cardiovascular stress test was performed using a cycle ergometer (Ergoline GmbH, Germany). During this test, electrocardiogram (ECG), heart rate, blood pressure, and well-being of the subject were monitored, and all complaints were documented. The load was started with 25 W, and it was gradually increased by 12.5 W every minute until the patient became exhausted or other criteria for study discontinuation were obvious: significant changes in ECG, hypotension (decrease in systolic BP > 20 mmHg), marked hypertension (> 230/120 mmHg), chest pain, pronounced shortness of breath, or other exercise-limiting symptoms. The intensity of the achieved load was evaluated in W and metabolic units (MET). One MET is the amount of oxygen while sitting consumed per kg of body weight in 1 min and equal to 3.5 ml/kg/min [20]. The cardiovascular stress test is the gold standard for the diagnostic evaluation of exercise intolerance especially in cardiac patients and for setting an individualized prescription of structured exercise training program [21;22;23]. This test is typically performed on a treadmill or stationary bike. The test shows very good-to-excellent validity (r = 0.85–0.97) and moderate-to-high reliability (intraclass correlation coefficient = 0.85–0.91) [24].”
Comments 17: Study interventions: Line 175: All participants were provided with a conventional physical therapy program. If all participants received a physiotherapy program, it should be more clearly stated whether the other three intervention programs were applied additionally. Response 17: Thank you for your comment. As we mentioned in the previous comments, we revised the Materials and methods section in order to make all information more clearly presented. The description of interventions given to different groups starts as follows: Patients were allocated to the following three groups: 1. Control group (CG) that underwent only conventional rehabilitation program (Table 1, part A); 2. Intervention group 1 (IG-1) that underwent the conventional rehabilitation program (Table 1, part A) plus multicomponent dynamic training (Table 1, part B); 3. Intervention group 2 (IG-2) that underwent the conventional rehabilitation program (Table 1, part A) plus the combined computer-based interactive cardiac pro-gram (Table 1, part C).
Later, we provide new Table 1 with the detailed description of each intervention.
Comments 18: Study interventions: Line 187: In addition, patients performed breathing and stretching exercises for 15–20 min, 7 days a week The fact that this exercise was given only to the control group creates ambiguity in the research design. It should be clarified why only the control group did these exercises and not the other groups. In addition, it should be stated in more detail how the effectiveness of this exercise was evaluated and on what criteria the results were obtained. Response 18: Thank you for your comment. We revised the Material and methods section as well as included a new table, and we hope that after this revision, everything will be clearer. Answering your question, stretching and breathing exercises were an integral part of the conventional rehabilitation program that was given to all study groups; therefore, there is no ambiguity regarding this aspect.
Comments 19: Discussion: The discussion section requires a more in-depth comparison of the research findings with the literature. The current discussion does not sufficiently relate the findings of the research to similar studies in the literature. The accuracy and meaning of the results obtained should be more clearly demonstrated by comparing with the existing literature. The parallels and differences between the findings obtained in similar studies in the literature and the results of the research should be emphasized, and the meaning of the findings, the factors that are important and how the results can be generalized should be emphasized. The main findings of the study are not clearly stated. It is expected that data will be presented on the main results of the study, such as differences between intervention groups, recovery rates, and changes in important parameters. Response 19: Thank you for your comment. We revised the entire discussion, and it appears as follows:
“4. Discussion This study aimed at evaluating and comparing the effects of different exercise interventions on fall risk and gait parameters among vulnerable and frail patients after open heart surgery during cardiac rehabilitation. The results of this study revealed positive and statistically significant results in reducing the risk of falls and improving gait parameters with additional interventions in frail patients after open heart surgery. Research shows that strength, power, and endurance trainings are the most used components for older adults [30]. It is widely known that multicomponent exercises are recommended strategy for frail older adults that increases functional capacity, balance, strength, gait speed, cognitive function and independence in the elderly [31;30]. We could not find any studies that used interactive technologies in patients with frailty, especially after open heart surgery. Studies conducted with interactive technologies in the field of frailty show that these exercises increase gait speed and balance [15]. Although the combined computer-based interactive cardiac program used in this study was superior compared to other programs, in the analysis of the literature, there is a lack of research on the effect of interactive training programs, therefore, more studies are required to validate the effects of interactive exercise training [15]. As a result, it was difficult to perform a comparable analysis with other research in this field, because we could not find similar studies investigating multicomponent and computer-based exercises programs, which were used together. Cadore et al. noted that multicomponent training is one of the most effective ways of improving physical fitness, gait ability, balance, and strength performance and reducing the rate of falls in frail individuals, especially when program consists of multiple components [32]. A study by Veronese et al. showed that individuals aged more than 65 years and having sarcopenia had a 1.85-fold greater likelihood of fall-related injury [33]; therefore, the inclusion one of the most important part – strength training – into a rehabilitation program could be a highly effective prevention strategy in delaying and mitigating the negative effects of sarcopenia and frailty in both early and late phases [34]. On the other hand, Khadanga et al. found that combination of aerobic exercise and strength training has been shown to be more effective in improving muscle strength, exercise performance, and cardiorespiratory fitness in frailty patients compared to using these trainings separately [35]. In our research multicomponent program included aerobic, sensorimotor, strength and flexibility training and this program decreased fall risk by 23.08%. Another systematic analysis presents slightly different statistics, however, they only examined multicomponent program effects. Daniel et al. revealed that in training programs that focused on multicomponent exercises, there was a 70% reduction in the risk of falls, a 54% increase in walking speed, an 80% improvement in balance, and a 70% increase in muscle strength among frail individuals [36]. Our study showed that multicomponent program improved fall risk outcome, otherwise patients who received the combined computer-based interactive cardiac program had a 15% and 22.73% reduced fall risk as assessed by overall stability index as compared with the patients in the group receiving multicomponent and in the control group receiving conventional rehabilitation. It is worth mentioning that before applying additional intervention programs, it is important to assess elderly patients’ frailty status as it can be a significant predictor of fall risk [37]. A recent meta-analysis performed in 2023, involving studies where frailty was defined by the usage of fall risk, demonstrated that computer-based interactive training increased lower limb muscle strength, walking speed, and balance as well as reduced fall risk. However, when frailty was defined by another frailty assessment model, i.e. Fried frailty phenotype criteria, computer-based interactive training improved only walking speed and balance [15]. To sum up, it is necessary to choose the appropriate assessment tool to evaluate frailty as this may influence the interpretation of fall risk results [3;15]. Many authors underpin the benefits of exercise interventions on reducing the risk of falling among elderly patients [38;4]; however, in a literature search there is an ongoing discussion what type, frequency, intensity, and duration of exercise interventions are most effective in the rehabilitation of frail patients after open heart surgery. Burton et al. found that multicomponent training as well as progressing intensity over time can contribute to decreasing the fall risk in older people [39]. The meta-analysis carried out in 2021 reported that an integrated intervention incorporating resistance, core, and balance training exceeding the duration of 32 weeks and frequency of 5 times a week was effective in reducing the risk of falling [38]. Even though in our study duration of multicomponent training was shorter (applied 3 times a week for 3 weeks) also resulted in significant changes in fall risk. Another review article showed that multicomponent training is associated with reduced fall risk, improved task performance and memory. Exercise intensity ranged from low, moderate and high, and the duration varied from 5 days to 6, 12, 16 and even 20 weeks [30]. This shows that not always a long duration can lead to a significant difference in the risk of falling. On the one hand, a fairly short time frame for the application of additional exercise interventions in our study allowed to document significant changes in fall risk, but on the other, we did not have any possibility to carry out this study in a long-term perspective. The meta-analysis by Schoberer et al. found that computer-based and balance training components had a positive effect on patient’s fall risk, and exercises done for more than 6 months were more effective than short-term ones. They noted surprising findings that additional long-term interventions causing physical overload could have a negative impact on fall risk especially among frail residents [40]. Theou et al. revealed that long-term multicomponent training with shorter-duration sessions (30–45 min) might be a better option for this population [41]. These studies have provided evidence that while applying additional intervention programs for frail people, it is essential to evaluate an overall duration depending on the type of intervention aiming to maximally improve patient’s physical outcomes. A narrative review published in 2022 reported that while applying computer-based exercises in the rehabilitation of frail adults older than 65 years, they were proven to successfully complement a conventional rehabilitation program, improve motor and cognitive functions, reduce fall risk, and possibly reverse frailty to prefrailty [17]. A systematic review conducted in 2020 concluded that before making any substantial recommendations on the inclusion of computer-based interventions in the rehabilitation of older adults, it is necessary to pay attention to several factors especially those that are associated with frailty [42]. All the above research agrees with our study showing that the application of multicomponent and computer-based interactive interventions in the rehabilitation of patients affected by frailty can improve health outcomes. Even though in our study, fall risk as assessed by the overall stability index was significantly reduced in all groups, when evaluating the obtained data, an appropriate method for frailty assessment and type of training according to term of rehabilitation should be taken into consideration to accurately interpret the results on fall risk. Summarizing, all these studies and reviews show that additional interventions for frail patients after open heart surgery are effective in reducing fall risk among elderly people, but incorporating additional interventions into a rehabilitation program requires the assessment patient’s frailty. As for gait parameters, in our study the greatest part of significant changes was documented when performing within-groups comparisons, and between-group comparisons revealed statistically significant differences mostly between the group receiving multicomponent training and the group that completed the combined computer-based interactive cardiac program. Ruiz-Ruiz et al. in 2021 reported that the most important parameters to be considered in the assessment of frail patients’ gait were double-support time, gait speed, stride time, step time, the number of steps per day or walking percentage per day [43]. As compared the findings of this study with the results of our study, all groups after rehabilitation in our study showed significant differences in double-stance phase, gait speed, stride time and step time parameters; however, the number of steps per day or walking percentage per day were not evaluated in our study. After the between-group comparison of changes in these gait parameters, we noticed that computer-based interactive training was most effective in improving double-stance phase, stride time, and step time parameters as compared with other rehabilitation tools. One of the review articles conducted in 2023 indicated that most studies used various exercise games for lower-limb muscle strength and balance, durations varied from 3 to 15 weeks and sessions were between 20 and 90 min [15]. Our study showed that the computer-based interactive cardiac program improved fall risk and gait parameters significantly when applied 20 days 3 times a week with training session duration of up to 60 min. Available evidence shows that patients affected by frailty mostly have reduced gait speed [44]. The research carried out in 2022 showed that slow gait speed is associated with poorer outcomes after cardiac surgery [45]. In the literature, most of the information is found on the increase in gait speed after multicomponent training interventions [31;46;47;48]. Simple gait speed measurements are often used in research [49;50]; however, the gold standard for gait analysis is computer-based systems using motion capture techniques and force platforms [51]. The results of our previous study, also including three independent groups, conducted in 2021 indicated that the intervention group receiving training with mechanical devices along with the conventional rehabilitation program showed the greatest improvement in gait speed, but the difference was not statistically significant as compared to other two groups [50]. In this current study, gait speed was also most increased in the computer-based interactive exercise intervention group, but compared to other groups statistically significant difference was not achieved. A comment by Kim and Oh has emphasized that in an accurate measurement of gait speed it is important to consider not only timing method (manual or automatic), but also the starting approach (standing or moving start) [52]. In our study, we employed a moving start approach as a protocol for the initiation of gait speed measurements; however, in such setting of measurements, the equipment is located against the walking direction, and this might interfere with regular walking. Moreover, the walking path must be about 10 m in length [52]. All these aspects of gait speed measurements, chosen timing and starting approach could have an influence and can affect determining a significant difference in gait speed. Thus, further research needs to be done. Some limitations of this study must be addressed. The study was carried out in a single center; therefore, the results cannot be applicable for a broader population. The limited timeframe of the study may not show the long-term impact of the interventions. There was a relatively small sample size and a high patient attrition rate. As vulnerable and mildly frail patients accounted for the largest part of this study population, a question arises if the conclusions can be applied to severely frail patients at the same extent as well. The inclusion criteria such as capability to walk more than 150 m based on the 6-minute walk test and open-heart surgery narrow the possible broader recruitment of participants as no frail patients who walked less than 150 m or underwent a stenting procedure were enrolled into our study. Kinesiophobia experienced after surgery or lack of skills in walking on the treadmill could have an impact on computerized gait assessment that could distort the real results.”
Comments 20: Discussion: Line 309: different physical therapy interventions "exercise interventions" "physical therapy interventions" should be decided. Response 20: Thank you for your comment. As we mentioned in our reply to comment 4, we changed the term "physical therapy interventions" to "exercise interventions" as it is more accurate and better reflects our study.
Comments 21: Discussion: Line 314: physical therapy programs Response 21: Thank you for your comment. As we mentioned in our reply to comment 4, we changed the term "physical therapy interventions" to "exercise interventions" as it is more accurate and better reflects our study.
Comments 22: Discussion: Line 315-318: Patients who received the combined computer based interactive cardiac program had a 15% and 22.73% reduced fall risk assessed by overall stability index as compared with the patients in the group receiving multicomponent dynamic aerobic balance and strength training and in the control group receiving conventional rehabilitation. These groups should be clearly defined in the method section. In the method section, there are groups as 1) The control group (CG), 2) The first intervention group (IG-1), 3) Patients randomized to the second intervention group (IG-2). In the method section, it should be stated which intervention groups received which programs and how these groups were formed. Response 22: Thank you for your comment. We added additional information in comments 8 and 13 to
Comments 23: Discussion: Line 320-330: This information includes a discussion of which findings of the study? Focusing on factors directly related to the content of the study and based only on the findings obtained in the discussion section would be a more scientific and systematic approach. When making a discussion based on the findings of the study, only data and literature related to the results of the study should be evaluated. Authors need to make a more direct discussion about the data they obtained. Literature information not directly related to the findings of the study may damage the clarity and focus of the study. In addition, after mentioning conditions such as osteoporosis, arthritis, and low back pain in this paragraph, moving on to risk factors such as gender creates a situation that disrupts the integrity of the subject and exhibits a scattered narrative. Response 23: Thank you for your comment. We revised the entire discussion.
Comments 24: Discussion: Line 331-332: is very scarce…Many authors… There is a contradiction in this paragraph. These two statements contradict each other, because if there is ‘very little’ literature, it would not be correct to use the expression ‘many authors’. Response 24: Thank you for your comment. We revised the entire discussion. No above-mentioned contradiction is left.
Comments 25: Discussion: Line 334-336: however, there is an ongoing discussion what type, frequency, intensity, and duration of physical therapy interventions are most effective in the rehabilitation 335 of frail patients after open heart surgery… This should be clearly emphasised in the methodology section of this study. intervention protocols should be addressed comprehensively in terms of frequency, duration and intensity. Response 25: Thank you for your comment. We revised the entire discussion.
Comments 26: Discussion: During the discussion of the research method, it is important to compare the applied intervention protocols with other research protocols in the existing literature. This comparison is necessary to evaluate the effectiveness of the interventions. Findings on the effectiveness of similar interventions in the literature should be presented and how they overlap or differ from the results of this study should be discussed. Response 26: Thank you for your comment. We revised the entire discussion.
|
||||||||||||||||||||||||||||||||||||||||||

Round 2
Reviewer 2 Report
Comments and Suggestions for Authors
The authors responded to my comments and questions and made corrections to the text of the manuscript. I have no other comments.
Reviewer 3 Report
Comments and Suggestions for Authors
I wish you success in your work.